# Limited efficacy of repeated praziquantel treatment in *Schistosoma mansoni* infections as revealed by highly accurate diagnostics, PCR and UCP-LF CAA (RePST trial)

Pytsje T. Hoekstra[1]*, Miriam Casacuberta-Partal[1], Lisette van Lieshout[1], Paul L. A. M. Corstjens[2], Roula Tsonaka[3], Rufin K. Assaré[4,5,6,7], Kigbafori D. Silué[4,5], Eliézer K. N'Goran[4,5], Yves K. N'Gbesso[8], Eric A. T. Brienen[1], Meta Roestenberg[1,9], Stefanie Knopp[6,7], Jürg Utzinger[6,7], Jean T. Coulibaly[4,5,6,7], Govert J. van Dam[1]

1 Department of Parasitology, Leiden University Medical Center, Leiden, The Netherlands, 2 Department of Cell and Chemical Biology, Leiden University Medical Center, Leiden, The Netherlands, 3 Department of Biomedical Data Sciences, Leiden University Medical Center, Leiden, The Netherlands, 4 Centre Suisse de Recherches Scientifiques en Côte d'Ivoire, Abidjan, Côte d'Ivoire, 5 Unité de Formation et de Recherche Biosciences, Université Félix Houphouët-Boigny, Abidjan, Côte d'Ivoire, 6 Swiss Tropical and Public Health Institute, Allschwil, Switzerland, 7 University of Basel, Basel, Switzerland, 8 Département d'Agboville, Centre de Santé Urbain d'Azaguié, Azaguié, Côte d'Ivoire, 9 Department of Infectious Diseases, Leiden University Medical Center, Leiden, The Netherlands

* pthoekstra@lumc.nl

## Abstract

### Background

Most studies assessing praziquantel (PZQ) efficacy have used relatively insensitive diagnostic methods, thereby overestimating cure rate (CR) and intensity reduction rate (IRR). To determine accurately PZQ efficacy, we employed more sensitive DNA and circulating antigen detection methods.

### Methodology

A sub-analysis was performed based on a previously published trial conducted in children from Côte d'Ivoire with a confirmed *Schistosoma mansoni* infection, who were randomly assigned to a standard (single dose of PZQ) or intense treatment group (4 repeated doses of PZQ at 2-week intervals). CR and IRR were estimated based on PCR detecting DNA in a single stool sample and the up-converting particle lateral flow (UCP-LF) test detecting circulating anodic antigen (CAA) in a single urine sample, and compared with traditional Kato-Katz (KK) and point-of-care circulating cathodic antigen (POC-CCA).

### Principal findings

Individuals positive by all diagnostic methods (i.e., KK, POC-CCA, PCR, and UCP-LF CAA) at baseline were included in the statistical analysis (n = 125). PCR showed a CR of 45% (95% confidence interval (CI) 32–59%) in the standard and 78% (95% CI 66–87%) in the intense treatment group, which is lower compared to the KK results (64%, 95% CI 52–75%) and 88%, 95% CI 78–93%). UCP-LF CAA showed a significantly lower CR in both groups,

**Data Availability Statement:** All relevant data are available within the manuscript and its supporting information file.

**Funding:** This work received financial support from the Prof. Dr. Flu Foundation, based in The Netherlands and the Coalition for Operational Research on Neglected Tropical Diseases (COR-NTD) (no. NTDSC087G) to GJD, which is funded at The Task Force for Global Health, primarily by the Bill & Melinda Gates Foundation, by the United States Agency for International Development through its Neglected Tropical Disease Program, and with UK Aid from the British people. The use of the PCR internal control PhHV was supported by the European Virus Archive goes Global (EVAg) project that has received funding from the European Union's Horizon 2020 research and innovation program under grant agreement No 653316. The funders had no role in the design of the study, data collection, analysis, decision to publish, or preparation of the manuscript.

**Competing interests:** The authors have declared that no competing interests exist.

16% (95% CI 11–24%) and 18% (95% CI 12–26%), even lower than observed by POC-CCA (31%, 95% CI 17–35% and 36%, 95% CI 26–47%). A substantial reduction in DNA and CAA-levels was observed after the first treatment, with no further decrease after additional treatment and no significant difference in IRR between treatment groups.

## Conclusion/Significance

The efficacy of (repeated) PZQ treatment was overestimated when using egg-based diagnostics (i.e. KK and PCR). Quantitative worm-based diagnostics (i.e. POC-CCA and UCP-LF CAA) revealed that active *Schistosoma* infections are still present despite multiple treatments. These results stress the need for using accurate diagnostic tools to monitor different PZQ treatment strategies, in particular when moving toward elimination of schistosomiasis.

## Clinical trial registration

www.clinicaltrials.gov, NCT02868385.

## Author summary

Efficacy of praziquantel (PZQ) for the treatment of schistosomiasis is usually assessed by classical microscopic detection of parasite eggs in stool or urine. Due to low sensitivity, especially in case of low-intensity infections, the prevalence of infection is underestimated leading to an overestimated cure rate (CR) when using these methods. In a repeated treatment trial, the efficacy of one versus four repeated PZQ treatments, given at 2-week intervals, was investigated in school-aged children from Côte d'Ivoire by applying a range of diagnostic methods, including traditional microscopy as well as more sensitive DNA and circulating antigen detection methods. Our results demonstrate that PZQ efficacy measurements vary based on the diagnostic method used: while egg-based diagnostics (stool microscopy and DNA detection methods) show an improved CR after repeated treatment, the CR determined by worm-based diagnostics (urine circulating antigen detection methods) remained poor over time. Although all four diagnostic methods showed a significant reduction in intensity of infection already after a single treatment, more accurate antigen diagnostics revealed that, in most cases, worms remain present even after multiple treatments. Hence, using accurate diagnostic tools is essential to determine the true infection status and to monitor and evaluate treatment programs.

## Introduction

Schistosomiasis remains a major public health problem in many sub-Saharan African countries. The cornerstone of schistosomiasis control programs is large-scale administration of the anthelmintic drug praziquantel (PZQ) [1]. Even though the World Health Organization (WHO) recommends using the intensity reduction rate (IRR) to measure the efficacy of PZQ treatment, the efficacy is still often expressed as cure rate (CR) [2]. Observed CRs in school-aged children range from 42% to 79% for *Schistosoma mansoni* and between 37% and 93% for *S. haematobium* (3). Repeating PZQ treatment can increase the CR up to 99% [3–5]. However, the majority of studies assessing the efficacy of PZQ treatment have used classical parasitologic methods, including Kato-Katz (KK) to determine the CR [6]. It is widely acknowledged that

these methods lack sensitivity, especially in case of low intensity infections–as often observed after treatment [7–9]–thereby underestimating the prevalence of infection, and hence, overestimating the CR [10–11].

To determine the efficacy of PZQ more accurately, diagnostic methods with a higher sensitivity should be used. Highly specific and sensitive molecular PCR techniques detecting *Schistosoma*-specific DNA in stool and urine are available and could be suitable for monitoring PZQ efficacy [7,12]. Alternatively, as PZQ affects the adult worm, treatment efficacy could also be evaluated by measuring the worm burden. Schistosome worms regurgitate various antigens into the blood circulation, which are then excreted via the urine, for example the antigens 'circulating cathodic antigen' (CCA) and 'circulating anodic antigen' (CAA); two antigens that have been described and studied extensively [13,14]. Detection of CCA is done via the commercially available point-of-care (POC) CCA urine cassette test, which is currently being recommended by WHO as an alternative for KK for diagnosing intestinal schistosomiasis [15–18]. CAA is detected quantitatively using an ultra-sensitive reporter technology (up-converting particle, UCP), combined with common immunochromatography, lateral flow (LF) [19]. This UCP-LF CAA test has demonstrated high sensitivity and specificity for the main human schistosome species (*S. haematobium*, *S. mansoni*, *S. japonicum*, and *S. mekongi*) [19–22]. Combining egg-derived nucleic acid and worm-derived antigen detection methods is expected to provide a more accurate determination of the efficacy of PZQ, both in terms of parasite worm dynamics as well as fecundity.

Previous results from the Repeated Praziquantel for Schistosomiasis Treatment (RePST) study–an open-label randomized controlled trial evaluating the efficacy of repeated PZQ on parasite clearance in school-aged children with a confirmed *S. mansoni* infection–showed a significantly higher CR after four PZQ treatments compared to a single PZQ based on the traditional KK technique [23]. However, the CR based on POC-CCA was substantially lower than the CR based on KK, even after four repeated PZQ treatments, indicating that worms are most likely still present after repeated treatment. Here, we further evaluate the efficacy of PZQ by using two additional sensitive and highly specific diagnostic methods, i.e., stool PCR and the UCP-LF CAA urine test, to determine the CR as well as the IRR after single and four repeated treatments. The current study uniquely combines all diagnostic methods, and provides new insights into the post-treatment dynamics of *S. mansoni* infections.

## Methods

### Ethics statement

Ethical approval was granted in Côte d'Ivoire from both the Comité National d'Éthique des Sciences de la Vie et de la Santé de Côte d'Ivoire (no. 091-18/MSHP/CNESVS-km) and the Direction de la Pharmacie, du Médicament et des Laboratoires de Côte d'Ivoire (no. 99433/MSPH/DGS/DPML/DAR and ECCI00618), and in The Netherlands from the Ethics Committee of the Leiden University Medical Center (P16.254). Oral assent from school-aged children as well as signed informed consent from children's parents or guardians was obtained before data and sample collection.

### Study design and participants

Previously, an open-label, randomized controlled trial was conducted between October 2018 and January 2019 in the Taabo health district in south-central Côte d'Ivoire, results of which have been published elsewhere [23,24]. Briefly, after clinical and parasitologic assessment for eligibility, school-aged children (5–17 years) with a confirmed *S. mansoni* infection (i.e., positive by KK (>8 eggs per gram of stool, EPG) and POC-CCA (traces excluded)), were

randomized into the 'standard treatment' or into the 'intense treatment' group [23]. At baseline, all participating children were given PZQ (40 mg/kg). Subsequently, children assigned to the standard group did not receive any further treatment during the trial period, while children assigned to the intense treatment group received additional PZQ treatment at 2, 4, and 6 weeks after the initial treatment [23].

## Diagnostic procedures

A detailed description of all field and laboratory procedures can be found in the previously published study protocol [24]. In brief, urine and stool samples were collected from each participating child weekly and two-weekly, respectively, over a period of 10 weeks in total. In the field, urine samples were subjected to the POC-CCA test using the G-score method [25], while stool samples were processed and examined using the KK technique [23,24]. Aliquots of all available urine and stool (mixed with ethanol for storage and transport purposes [26]) samples were stored at -20˚C. After the trial was completed, all available urine and stool aliquots were shipped to the Leiden University Medical Center (Leiden, The Netherlands) and stored at -20˚C pending further testing.

The *Schistosoma* genus specific (ITS2) real-time PCR was used for the detection of *Schistosoma* DNA in stool samples, as described previously [27,28]. Besides negative and positive control samples included in each amplification run, phocin herpes virus-1 (PhHV-1) was added to the lysis buffer in each sample as an internal positive control. Virus-specific primers and detecting probe were included in each reaction mixture. Fifty PCR amplification cycles were run per sample, using the amplification cycle in which the level of fluorescent signal exceeded the background as the cycle-threshold (Ct)-value PCR output. Since its implementation, the LUMC-team scored 100% in sensitivity and specificity of their *Schistosoma* PCR at the annual international Helminths External Molecular Assessment Scheme (HEMQAS) provided by the Dutch Foundation for Quality Assessment in Medical Laboratories (SKML) [29].

Urine samples were subjected to the UCP-LF CAA test [19]. A set of reference standards with a known CAA concentration were included to quantify CAA-levels as well as to validate the cut-off (0.6 pg/ml for the urine UCAA*hT*417 test [19]). Samples were considered positive if the CAA concentration exceeded the cut-off, while samples below the cut-off were considered negative.

## Statistical analysis

Only individuals with a positive baseline outcome in all diagnostic methods (i.e., KK, POC-CCA, PCR, and UCP-LF CAA) were included in the analysis. As the diagnostics used typically focus on direct detection of the presence of worms (by their metabolic excretion products, CCA and CAA) or indirectly by showing the presence of the eggs (KK and PCR), data were analyzed to compare both approaches. Descriptive statistics were performed using IBM Statistical Package for Social Sciences version 25 (SPSS Inc., Chicago, United States of America). Prevalence over time as well as CR and IRR based on PCR and UCP-LF CAA were determined using a mixed effects model taking into account the correlation between the different measurements from the same individual [30–32]. CR and IRR were determined by comparing baseline data to outcomes at 4 weeks after the last PZQ treatment, i.e., considering week 4 as the final time point for the standard treatment group and week 10 as the final time point for the intense treatment group. Ct-values were transformed into arbitrary units (AU) of copy numbers of *Schistosoma* DNA, as described before [33]. In short, to each low positive sample, i.e., a Ct-value between 35 and 50, an arbitrary value of 1 AU was assigned. Starting from Ct-value 35, for each PCR cycle reduction a duplication of AU was computed, assuming 100% efficacy of the DNA multiplication process.

To determine the prevalence over time as well as the CR and IRR (based on all diagnostic methods) mixed effects models were used, as described previously [23,31]. This model framework takes into account the zero inflation, the correlation between repeated measurements from the same individual and gives valid results under the missing at random assumption for the missing data, which is valid in this study. For the prevalence and CR, we used mixed effects logistic regression where prevalence was modeled as a function of time, treatment group, and their interaction. In the case of KK and PCR, the time variable was taken as categorical, while for POC-CCA and UCP-LF CAA we modeled progression over time using natural cubic splines with knots. For the IRR, the mean was modeled using the main effect of time (taken as categorical), the main effect of treatment and their interaction. KK data were analyzed in time using a zero-inflated mixed effects negative binomial model, while PCR and UCP-LF CAA data were analyzed in time using a two-part linear mixed model. The correlation between the repeated measurements of each individual was modeled using random effects (i.e., random intercepts). All analyses have been done in R (version 3.43) using the GLMMadaptive package. CR and IRR estimated from the model are given with their corresponding 95% pointwise confidence intervals (CIs).

In the absence of a true 'gold' standard, the sensitivity of the different diagnostic methods was determined by comparison against a composite reference standard (CRS) [34–36], which was based on a positive result by KK and/or PCR and/or UCP-LF CAA, all considered highly specific methods. Consequently, an individual was considered true positive if either KK, PCR, or UCP-LF CAA was positive.

## Results

From the 153 school-aged children who participated in the original trial, in 19 cases no aliquots from baseline urine and/or stool samples were available for additional testing. Another nine school-aged children were omitted as they had a negative UCP-LF CAA (n = 7) or a negative PCR (n = 2) at baseline. In total, 125 school-aged children were included in the final analysis, 56 and 69 in the standard and intense treatment group, respectively (S1 Fig). The demographic and parasitologic baseline data for the school-aged children included in the current analysis are summarized in S1 Table.

### Prevalence over time

In Fig 1, the percentage of *S. mansoni* positive samples over time based on PCR (Fig 1A) and UCP-LF CAA (Fig 1B) is shown, including previously published results based on KK and POC-CCA [23] (Fig 1C and 1D, respectively). Based on PCR, the total percentage of positives decreased to 55% (95% CI 41–68%) in the standard treatment group and to 22% (95% CI 13–34%) in the intense treatment group, both measured 4 weeks post-treatment. The corresponding CR for the standard and intense treatment group were 45% (95% CI 32–59%) and 78% (95% CI 66–87%), respectively (Tables 1 and S2). Based on UCP-LF CAA, the total percentage of positives decreased to 86% (95% CI 79–90%) in the standard treatment group and to 82% (95% CI 73–89%) in the intense treatment group, both measured 4 weeks post-treatment. The corresponding CR for the standard and intense treatment group were 16% (95% CI 11–24%) and 18% (95% CI 12–26%), respectively (Tables 1 and S2).

### Intensity of infection over time

Fig 2 illustrates the infection intensity over time based on PCR (Fig 2A) and UCP-LF CAA (Fig 2B), including KK and POC-CCA results (Fig 2C and 2D, respectively). Based on PCR, most remaining infections after the first PZQ treatment were of low intensity. In the standard

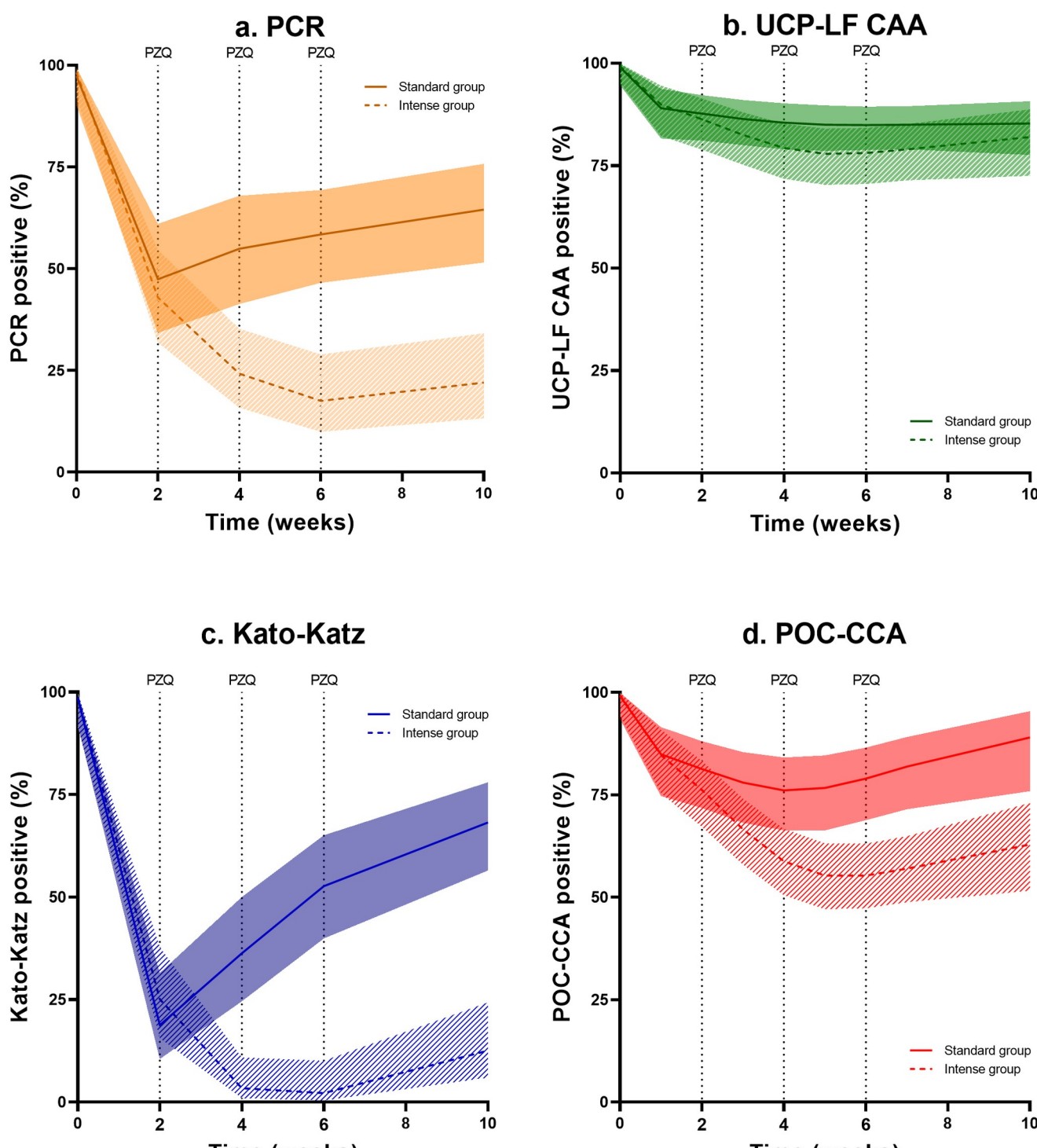

**Fig 1. Percentage positive over time with corresponding 95% confidence intervals estimated from the mixed effects logistic regression model.** Data based on (a) polymerase chain reaction (PCR), (b) up-converting particle circulating anodic antigen (UCP-LF CAA), (c) Kato-Katz (KK), and (d) point-of-care circulating cathodic antigen (POC-CCA), in the standard treatment group (n = 56, single dose of PZQ at week 0, solid line) and intense treatment group (n = 69, four doses of PZQ at weeks 0, 2, 4, and 6, dashed line).

**Table 1. Cure rate (CR) and intensity reduction rate (IRR) of a single (standard treatment group) and four (intense treatment group) repeated PZQ treatments in 125 school-aged children with a confirmed _S. mansoni_ infection.** Data based on polymerase chain reaction (PCR) and up-converting particle circulating anodic antigen (UCP-LF CAA).

|  | Standard treatment group (n = 56) | Intense treatment group (n = 69) |
|---|---|---|
| **PCR** | | |
| Cured children 4 weeks post-treatment | 25 | 52 |
| CR (95% CI)[a] | 45.3% (32.3–58.8%) | 78.1% (66.4–86.6%) |
| Median AU[b] | | |
| Before treatment | 32,768 | 16,384 |
| 4 weeks post-treatment | 2,048 | 24 |
| Arithmetic mean AU[a] | | |
| Before treatment (95% CI) | $5.6 \times 10^5$ ($2.3 \times 10^5$–$1.2 \times 10^6$) | $2.0 \times 10^5$ ($9.5 \times 10^4$–$4.2 \times 10^5$) |
| 4 weeks post-treatment (95% CI) | $1.0 \times 10^4$ ($3.4 \times 10^3$–$2.8 \times 10^4$) | $9.9 \times 10^2$ ($2.2 \times 10^2$–$4.3 \times 10^3$) |
| IRR (95% CI)[a] | 99.6% (98.6–99.9%) | 99.5% (97.2–99.9%) |
| **UCP-LF CAA** | | |
| Cured children 4 weeks post-treatment | 9 | 13 |
| CR (95% CI)[a] | 16.1% (10.5–24.0%) | 17.8% (11.5–26.3%) |
| Median urine CAA-level (pg/ml)[b] | | |
| Before treatment | 286 | 270 |
| 4 weeks post-treatment | 61 | 14 |
| Arithmetic mean urine CAA-level (pg/ml)[a] | | |
| Before treatment (95% CI) | 145.6 (93.3–217.8) | 153.4 (102.5–219.3) |
| 4 weeks post-treatment (95% CI) | 40.6 (25.2–62.2) | 15.2 (9.3–23.3) |
| IRR (95% CI)[a] | 72.1% (62.4–79.4%) | 90.1% (86.9–92.5%) |

Abbreviations: AU, arbitrary unit (see Methods for definition); CAA, circulating anodic antigen; CR, cure rate; IRR, intensity reduction rate; PCR, polymerase chain reaction; PZQ, praziquantel; UCP-LF, up-converting particle lateral flow.

a Calculated from the model

b Median of the positives

treatment group, the average PCR outcome (AU) decreased from $56 \times 10^4$ (95% CI $23 \times 10^4$–$12 \times 10^5$) to $10 \times 10^3$ (95% CI $34 \times 10^2$–$28 \times 10^3$), and in the intense treatment group from $20 \times 10^4$ (95% CI $95 \times 10^3$–$42 \times 10^4$) to $99 \times 10^1$ (95% CI $22 \times 10^1$–$43 \times 10^2$), both measured 4 weeks post-treatment. The corresponding IRR for the standard and intense treatment group were 99.6% (95% CI 98.6–99.9%) and 99.5% (95% CI 97.2–99.9%), respectively (Tables 1 and S2). Based on UCP-LF CAA, infection levels decreased from 146 pg/ml (95% CI 93–218) to 41 pg/ml (95% CI 25–62) in the standard treatment group, and from 153 pg/ml (95% CI 103–219) to 15 pg/ml (95% CI 9–23) in intense treatment group. The corresponding IRRs for the standard and the intense treatment group were 72.1% (95% CI 62.4–79.4%) and 90.1% (95% 86.9–92.5%), respectively (Tables 1 and S2).

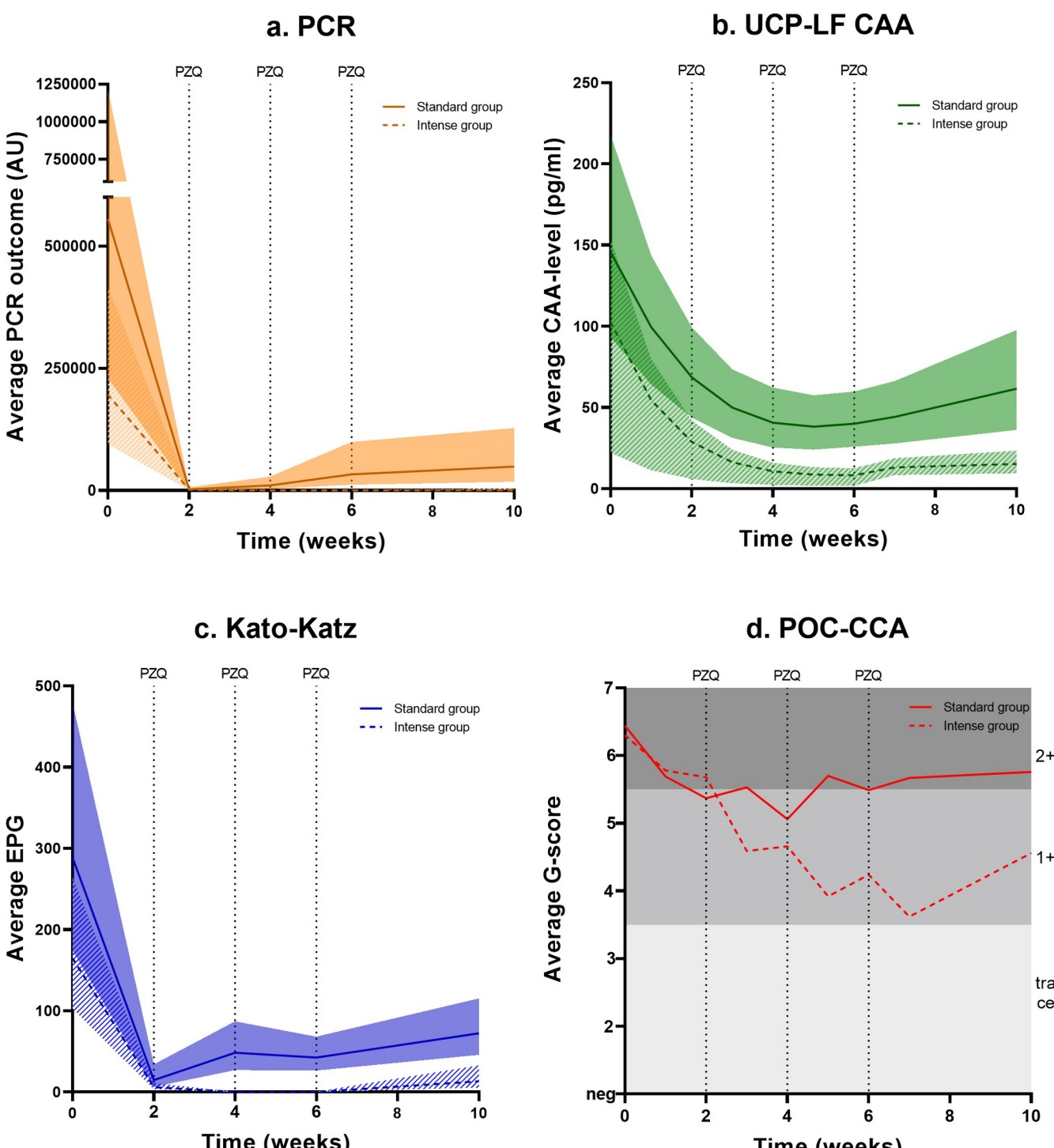

**Fig 2. Average infection levels over time with corresponding 95% confidence intervals* estimated from the mixed effects logistic regression model.** Data shown for (a) polymerase chain reaction (PCR), (b) up-converting particle circulating anodic antigen (UCP-LF CAA), (c) Kato-Katz (KK), and (d) point-of-care circulating cathodic antigen (POC-CCA), in the standard treatment group (n = 56, single dose of PZQ at week 0, solid line) and intense treatment group (n = 69, four doses of PZQ at weeks 0, 2, 4, and 6, dashed line). * for POC-CCA no confidence intervals available.

## a. Egg-based detection methods

## b. Worm-based detection methods

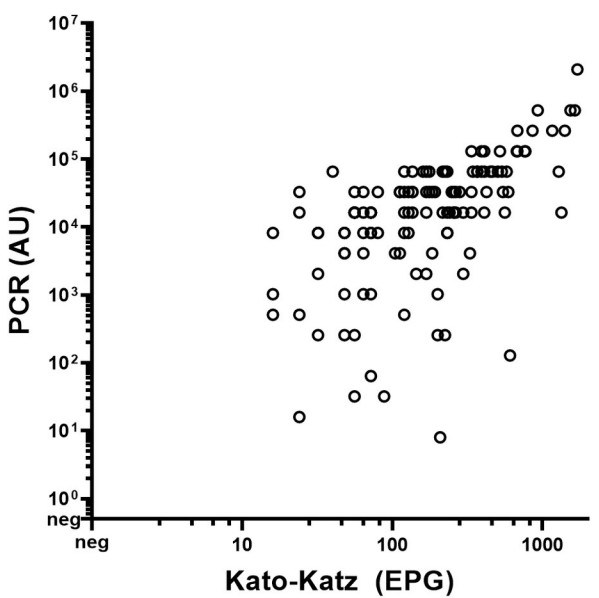
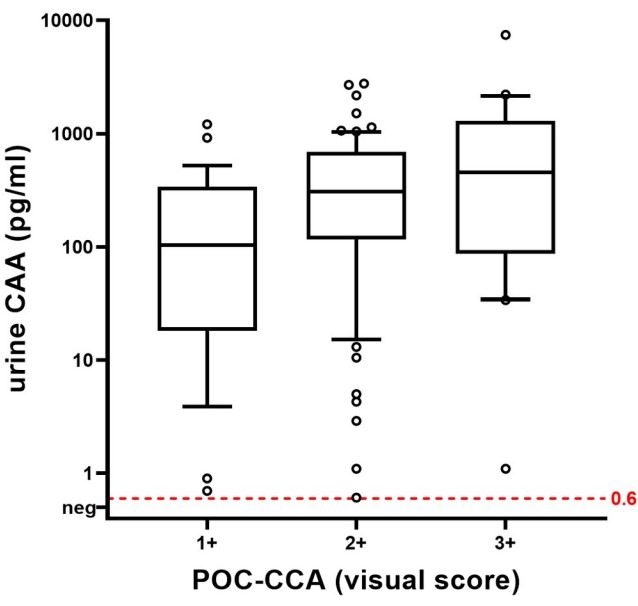

**Fig 3. Baseline correlation between (a) egg-detection methods and (b) worm-based detection methods.** Data shown for (a) polymerase chain reaction (PCR) versus Kato-Katz (KK) and (b) point-of-care circulating cathodic antigen (POC-CCA) versus up-converting particle circulating anodic antigen (UCP-LF CAA) at baseline (n = 125).

### Correlation between tests

In Fig 3, the baseline correlation between egg-based detection methods (Fig 3A) and worm-based detection methods (Fig 3B) is shown. The correlation between egg-based methods PCR (in AU) and KK (in EPG) at baseline was higher (Spearman's rho 0.64, p<0.01) compared to the correlation between worm-based detection by UCP-LF CAA and POC-CCA (Spearman's rho 0.37, p<0.01). The majority of individuals with ≥400 EPG (86%) had a POC-CCA G-score of 6 or higher (corresponding to a visual score of 2+ or higher) and a urine CAA-level of ≥100 pg/ml (S2 and S3 Figs).

Fig 4 shows the post-treatment correlation between egg-based detection methods (Fig 4A) and worm-based detection methods (Fig 4B). A similar correlation was observed between PCR and KK (Spearman's rho 0.51, p<0.01) and between UCP-LF CAA and POC-CCA (Spearman's rho 0.59, p<0.01) 4 weeks after treatment. The majority of egg- and PCR-positive individuals were observed in the standard treatment group, while UCP-LF CAA and POC-CCA positives were observed in both groups (S4 Fig).

### Accuracy of tests

Sensitivity of egg-based diagnostic methods decreased over time after treatment; a higher reduction in sensitivity was observed in the intense treatment group compared to the standard treatment group (Table 2). While in both groups the sensitivity of POC-CCA fluctuated over time, the sensitivity of UCP-LF CAA remained high and stable, regardless of additional treatment.

### Discussion

In this study, performing additional diagnostic analysis on samples previously collected at a multiple treatment trial, comparing single versus 4 treatments with PZQ in school-aged

## a. Egg-based detection methods

## b. Worm-based detection methods

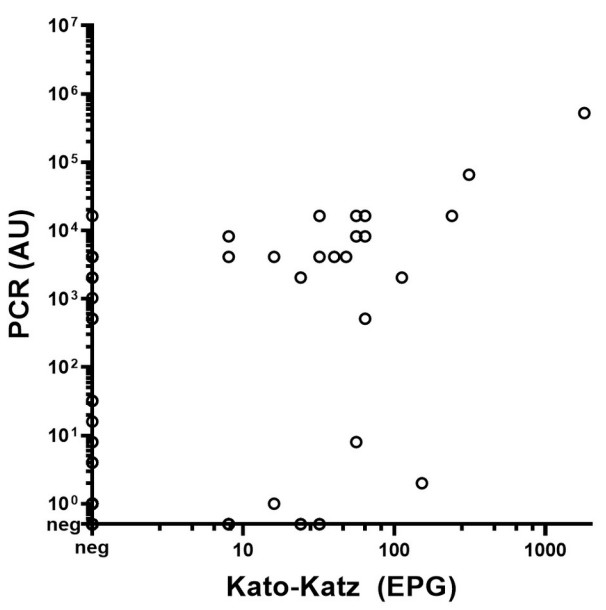
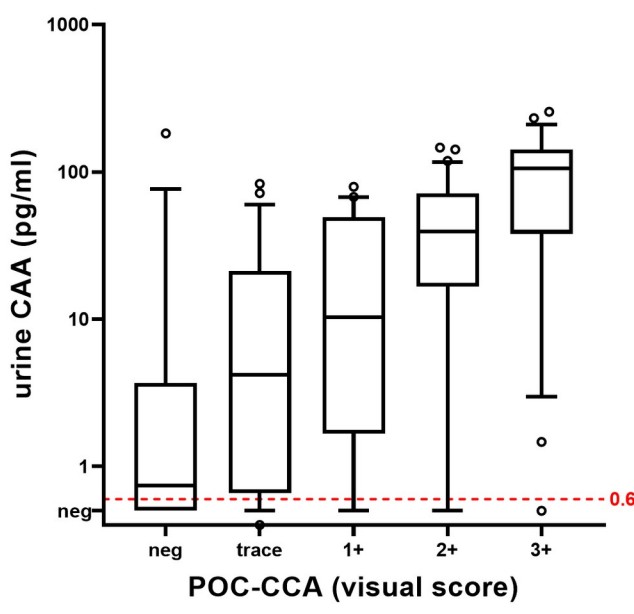

**Fig 4. Post-treatment correlation between (a) egg-detection methods and (b) worm-based detection methods.** Data based on (a) polymerase chain reaction (PCR) versus Kato-Katz (KK) and (b) point-of-care circulating cathodic antigen (POC-CCA) versus up-converting particle circulating anodic antigen (UCP-LF CAA) 4 weeks after treatment (n = 125).

children from Côte d'Ivoire, we confirmed our prior conclusion that highly accurate diagnostic methods are important for determining PZQ efficacy [23].

Due to its high sensitivity compared to KK, while remaining intrinsically specific, the *Schistosoma* genus-specific ITS2 PCR detecting *Schistosoma* DNA in stool samples revealed more positive cases over time. Yet, the overall post-treatment dynamics were similar to KK results [23]. This supports the notion that both tests reflect the number of *Schistosoma* eggs in stool

**Table 2. Sensitivity of polymerase chain reaction (PCR), Kato-Katz (KK), up-converting particle circulating anodic antigen (UCP-LF CAA), and point-of-care circulating cathodic antigen (POC-CCA) compared to a composite reference standard (CRS) over time.** Data shown for the standard group (n = 56) who received a single PZQ treatment at week 0 and the intense treatment group (n = 69) who received four repeated PZQ treatments at weeks 0, 2, 4, and 6. The composite reference standard (CRS) was based on KK, PCR, and UCP-LF CAA: an individual was considered positive if at least one of these tests was positive (i.e., test specificity assumed to be 100%).

| | | | | Number of individuals positive per diagnostic test and corresponding sensitivity (%) compared to the composite reference standard (CRS) | | | | | | | |
|---|---|---|---|---|---|---|---|---|---|---|---|
| | | **N** | **CRS** | **Stool PCR** | | **Kato-Katz** | | **UCP-LF CAA** | | **POC-CCA** | |
| **Standard group** | Baseline | 56 | 56 | 56 | 100% | 56 | 100% | 56 | 100% | 56 | 100% |
| | Week 2 | 54 | 48 | 25 | 52.0% | 10 | 20.8% | 46 | 95.8% | 42 | 87.5% |
| | Week 4 | 55 | 50 | 30 | 60.0% | 20 | 40.0% | 46 | 92.0% | 38 | 76.0% |
| | Week 6 | 55 | 49 | 32 | 65.3% | 29 | 59.2% | 46 | 93.9% | 43 | 87.8% |
| | Week 10 | 53 | 51 | 34 | 66.7% | 36 | 70.6% | 44 | 86.3% | 45 | 88.2% |
| **Intense group** | Baseline | 69 | 69 | 69 | 100% | 69 | 100% | 69 | 100% | 69 | 100% |
| | Week 2 | 68 | 61 | 29 | 47.5% | 17 | 27.9% | 58 | 95.1% | 51 | 83.6% |
| | Week 4 | 69 | 57 | 16 | 28.1% | 2 | 3.5% | 55 | 96.5% | 42 | 73.7% |
| | Week 6 | 63 | 49 | 11 | 22.4% | 1 | 2.0% | 46 | 93.9% | 34 | 69.4% |
| | Week 10 | 66 | 55 | 14 | 25.5% | 8 | 14.5% | 54 | 98.2% | 40 | 72.7% |

(as adult *Schistosoma* worms are located in the mesenteric blood vessels), although the ITS2 PCR target is considered to be present in all life-stages of the parasite. The good correlation between KK and PCR is consistent with previous studies [12,27,28,37,38] indicating the capability of the *Schistosoma* ITS2 PCR to determine infection intensity. PCR also showed a significant reduction in DNA levels after treatment, as observed previously [12,38], confirming its usefulness to monitor PZQ treatment efficacy. The latter is in contrast to other studies using a serum-based PCR that remained positive for months after treatment [39,40].

Turning to worm-based diagnostics, the highly accurate UCP-LF CAA test confirmed the already low CR observed previously by POC-CCA [23]. In 285 out of 483 (59%) post-treatment samples, CAA was detected in urine, while no eggs were observed in the stool. To a lesser extent, this was already shown by previously published POC-CCA positive but egg-negative individuals [23,41,42]. As both CAA and CCA are worm-derived antigens, the presence of either antigen indicates that worms are still metabolically active and that the infection is hence not fully cleared, indicating that the CR is even lower than generally assumed [3,6,43]. The opposite, egg-positive with no CAA in urine, was observed in only a few cases (15 out of 483, 3%). This apparent incongruence of absence of CAA despite presence of eggs could be due to variation in biological excretion, but administrative errors such as sample processing and/or labelling cannot be excluded.

Egg-based diagnostic methods (i.e., PCR and KK), showed an increased CR after repeated treatment, as also observed in other studies [3–5]. Contrastingly, the more sensitive worm-based methods, UCP-LF CAA and POC-CCA, demonstrated a poor CR, which did not significantly improve with additional treatments. Observed CRs should therefore be interpreted with care, keeping in mind that they depend on the type of diagnostic method used. In addition, the time point after treatment chosen for measurement of CRs also influences its value, e.g., for KK it is highest 2 weeks after treatment, and subsequently decreases in the next weeks if no additional treatment is given (S5A Fig). Using different diagnostics, our results confirm that the CR has limited value in assessing PZQ efficacy. Diagnostic sensitivity has a higher impact on the CR than on the IRR, as the detection of an egg, DNA, or circulating antigen makes the difference between cured and non-cured (S5 and S6 Figs).

A significant decrease in urine CAA-levels was observed already after a single treatment and no substantial differences were observed whether data were calculated as arithmetic or geometric mean or median (S7 Fig). By repeating the treatments at relatively short intervals, it was anticipated that non-PZQ-susceptible schistosomula would be targeted more effectively as with a few weeks they would have matured into PZQ-susceptible worms. Also, an additional effect on mature worms was expected, as the short metabolic half-life of PZQ might limit its effectiveness when given as single treatment. Our results indicate that even though a further reduction in egg output occurred with multiple treatments, worm numbers reflected by urine CAA-levels continued to decrease slightly and remained solidly low from 3 weeks onwards, with only very few cases becoming negative. The rapid rise in egg numbers after treatment would indicate that in this setting transmission continued which, for CAA, resulted in continuous low levels originating probably from young worms (2–4 weeks old) [14,19]. Alternatively or additionally, worms could reside in capillaries not reached by a single dose of PZQ, or worms could revitalize after the damage occurred by PZQ and the host immune attack, thus surviving the treatment. This would suggest that repeating treatment at 2-week intervals is not sufficient and that shorter intervals or perhaps a higher dose could be considered. Although generally rapidly cleared, it cannot be excluded that also a slow release of antigens, restrained in various tissues, contributed to the low CAA-levels after treatment. To further elucidate these mechanisms and to differentiate between surviving adult worms and establishing new infections (reinfection), research in endemic settings with various (seasonal) transmission

patterns, in particular the absence of transmission is needed. Alternatively, experimental infections in animals or controlled human infection models would be helpful in unravelling the exact metabolic aspects of CAA excretion and immune-mediated clearance patterns. Due to limited sensitivity of KK for light intensity infections after treatment, eggs may also continue to be excreted but remain below the detection limit of microscopy. Eggs that may be trapped in the host tissue, and therefore not detected, could still result in a continued risk of pathology [44]. Further research is needed to determine to the impact of the continued presence of worms without the detection of eggs and to what extent the egg production and excretion is reconstituted over time.

## Conclusion

A single PZQ treatment had a major impact on the egg output as well as on the worm burden measured by CAA-levels, but it appears difficult to get rid of the remaining worms. Even though additional treatment resulted in a further decrease in the number of eggs, the effect on the number of worms remaining in the host was limited, which indicates that–in this setting–(up to) four repeated treatments at a relatively short interval were not sufficient to clear all *S. mansoni* infections. In particular when moving toward interruption of transmission or even elimination of schistosomiasis, it is important to focus on complete clearance of infection, i.e., not only the absence of eggs, but more importantly the absence of worms. To determine the ideal schistosomiasis treatment schedule, still considering other important factors influencing post-treatment status such as transmission season, environmental conditions, and water contact behavior, the use of accurate egg- and worm-based diagnostic tools is essential.

## Supporting information

**S1 Fig. Trial profile.**
(PNG)

**S2 Fig. Correlation between worm-based detection methods including intensity categories based on Kato-Katz (eggs per gram of feces, EPG) at (a) baseline and (b) 4 weeks after (the last) treatment.** Data based on point-of-care circulating cathodic antigen (POC-CCA) and up-converting particle circulating anodic antigen (UCP-LF CAA) in combination with Kato-Katz (KK), with colors indicating the EPG intensity (n = 125).
(TIF)

**S3 Fig. Baseline correlation between egg-detection methods and worm-based detection methods.** Data based on stool polymerase chain reaction (PCR) versus Kato-Katz (KK) (a-b) and point-of-care circulating cathodic antigen (POC-CCA) versus up-converting particle circulating anodic antigen (UCP-LF CAA) (c-f) at baseline (n = 125).
(TIF)

**S4 Fig. Post-treatment correlation between egg-detection methods and worm-based detection methods.** Data based on stool polymerase chain reaction (PCR) versus Kato-Katz (KK) (a-b) and point-of-care circulating cathodic antigen (POC-CCA) versus up-converting particle circulating anodic antigen (UCP-LF CAA) (c-f) 4 weeks post-treatment (n = 125).
(TIF)

**S5 Fig. Cure rate (CR) over time determined by the different diagnostics in (a) the standard treatment group, who received a single dose of PZQ at week 0, and (b) the intense treatment group, who received four doses of PZQ at weeks 0, 2, 4, and 6.** Data based on stool polymerase chain reaction (PCR), Kato-Katz (KK), urine up-converting particle

circulating anodic antigen (UCP-LF CAA), and point-of-care circulating cathodic antigen (POC-CCA) (n = 125).
(TIF)

**S6 Fig. Intensity reduction rate (IRR) over time determined by the different diagnostics in (a) the standard group, who received a single dose of PZQ at week 0, and (b) the intense group, who received four doses of PZQ at weeks 0, 2, 4, and 6.** Data based on stool polymerase chain reaction (PCR), Kato-Katz (KK), and urine up-converting particle circulating anodic antigen (UCP-LF CAA) (n = 125).
(TIF)

**S7 Fig. Circulating anodic antigen (CAA) levels over time in (a) the standard group, who received a single dose of PZQ at week 0, and (b) the intense group, who received four doses of PZQ at weeks 0, 2, 4, and 6.** Data shown as arithmetic mean (AM), arithmetic mean of the positives, geometric mean (GM) of the positives, and median.
(TIF)

**S1 Table. Baseline characteristics of the standard treatment group and the intense treatment group.** Adapted from (24), for the current study based on data from 125 school-aged children.
(DOCX)

**S2 Table. Cure rate (CR) and intensity reduction rate (IRR) of a single (standard treatment group) and four (intense treatment group) repeated PZQ treatments in 125 school-aged children with a confirmed *S. mansoni* infection.** Data based on polymerase chain reaction (PCR), Kato-Katz (KK), up-converting particle circulating anodic antigen (UCP-LF CAA), and point-of-care circulating cathodic antigen (POC-CCA).
(DOCX)

## Acknowledgments

We would like to thank E. van Kaathoven for her support in performing the UCP-LF CAA and PCR assays. We thank all children for their enthusiastic participation and providing samples. We are grateful to the parents and guardians of children and the communities for allowing us to conduct this trial. We thank the technicians, nurses, volunteers, and drivers who assisted during the field and laboratory work. Special thanks are addressed to the directors and staff of Centre Suisse de Recherches Scientifiques en Côte d'Ivoire and the 'Programme National de Lutte contre les Maladies Tropicales Négligées à Chimiothérapie Préventive' for administrative support.

## Author Contributions

**Conceptualization:** Pytsje T. Hoekstra, Lisette van Lieshout, Meta Roestenberg, Stefanie Knopp, Jürg Utzinger, Jean T. Coulibaly, Govert J. van Dam.

**Data curation:** Pytsje T. Hoekstra, Miriam Casacuberta-Partal, Roula Tsonaka.

**Formal analysis:** Pytsje T. Hoekstra, Roula Tsonaka.

**Funding acquisition:** Govert J. van Dam.

**Investigation:** Pytsje T. Hoekstra, Miriam Casacuberta-Partal, Rufin K. Assaré, Kigbafori D. Silué, Yves K. N'Gbesso, Eric A. T. Brienen, Jean T. Coulibaly, Govert J. van Dam.

**Methodology:** Pytsje T. Hoekstra, Meta Roestenberg, Stefanie Knopp, Jürg Utzinger, Jean T. Coulibaly, Govert J. van Dam.

**Project administration:** Pytsje T. Hoekstra, Miriam Casacuberta-Partal, Rufin K. Assaré, Kigbafori D. Silué, Jean T. Coulibaly.

**Resources:** Pytsje T. Hoekstra, Miriam Casacuberta-Partal, Rufin K. Assaré, Kigbafori D. Silué, Eliézer K. N'Goran, Jean T. Coulibaly, Govert J. van Dam.

**Supervision:** Lisette van Lieshout, Govert J. van Dam.

**Validation:** Pytsje T. Hoekstra, Govert J. van Dam.

**Visualization:** Pytsje T. Hoekstra.

**Writing – original draft:** Pytsje T. Hoekstra.

**Writing – review & editing:** Pytsje T. Hoekstra, Miriam Casacuberta-Partal, Lisette van Lieshout, Paul L. A. M. Corstjens, Roula Tsonaka, Rufin K. Assaré, Kigbafori D. Silué, Eric A. T. Brienen, Meta Roestenberg, Stefanie Knopp, Jürg Utzinger, Jean T. Coulibaly, Govert J. van Dam.

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
