## [Decision Letter · Decision Letter 0]

9 Oct 2022

Dear Mrs Hoekstra,

Thank you very much for submitting your manuscript "Limited efficacy of repeated praziquantel treatment in Schistosoma mansoni infections as revealed by highly accurate diagnostics, PCR and UCP-LF CAA (RePST trial)" for consideration at PLOS Neglected Tropical Diseases. As with all papers reviewed by the journal, your manuscript was reviewed by members of the editorial board and by several independent reviewers. In light of the reviews (below this email), we would like to invite the resubmission of a significantly-revised version that takes into account the reviewers' comments. 

Dear Dr Hoekstra,

Firstly, I'd like to apologise for the lengthy delay in processing your manuscript. I trust the journal has been in contact with you. This delay was in part due to, I think, major changes in the journals submission cite, which meant the article sat in limbo for some weeks before it actually appeared in my assignment feed. I then had some further significant delays with finding reviewers. I mention this not to make excuses for these delay but by way of acknowledging that the length of time it has taken to process your submission is not acceptable and I hope it will not be repeated in the future.

As you'll see from the reviewers comments, the reviews are quite split on your manuscript. Unfortunately, somehow, one of the reviewer's (who recommended rejection of the manuscript) comments were not uploaded or otherwise went missing. I can't say what has caused this, but I followed up with the review and they provided the following summary of their concerns.

"The manuscript "Limited efficacy of repeated praziquantel treatment in Schistosoma mansoni infections as revealed by highly accurate diagnostics, PCR and UCP-LF CAA (RePST trial)" aimed to evaluate PZQ efficacy using PCR and UCP-LF CAA in addition to the techniques already employed in this kind of study (KK and POC-CCA). However, the results did not show the relevance of the different methods used. Also, there is no relevant difference between the obtained data from KK and POC-CCA and PCR and UCP-LF CAA. Accordingly, I recommend rejecting the manuscript."

In an effort to prevent any further delays to processing your submission, I have informally reviewed the article myself, particularly in order to see if I could understand this second reviewer's concerns. I think it would be helpful to see the comparisons you undertook in your supplementaries featuring more prominently in your manuscript in order to address these concerns. I certainly won't tell you how to design your study or author your manuscript, but I think it might be helpful if you considered a main text figure and/or table (both perhaps?) clearly demonstrating:

1. The differences in sensitivity of the four methods you employed and their relative agreement with your surrogate 'gold-standard'. You mention in your text that the UPC-LC CAA method has been shown in prior study to have 'high' specificity and sensitivity, but I think demonstrating this using your control material is important for reader's in interpreting the PZQ CR results. I understand that this is in your supplementaries, but I think it might be better placed in the main text.

2. Similarly, you demonstrate in the supplementaries, as notable difference in the estimated CR based on the four methods you have used, but again, I think having this as a main figure and highlighting which of these differences are statistically significant would be helpful to the reader.

2. Finally, do you have data for the relative diagnostic specificity of each method? Particularly specificity for patent/live infections. With the antigen tests, is there any way to know for how long hosts will continue to shed worm antigens after the worms themselves have been killed through PZQ treatment? Similarly, is there any data that following the apparent cessation of egg production after PZQ (inferred by KK for example) that hosts still showing infection based on UPC-LC CAA or POC-CCA eventually resume egg-shedding (as evidence of the survival of any remaining worms and their return to reproductive fecundity). I presume such data would have to come from a hamster or similar model animal. I think this is particularly worth considering in the intense treatment group in whom you measured a substantial reduction in infection based on KK, but much less so with the other methods after week 2 or so. Could it be that either treatment has sterilised the surviving adults (as has been seen in, for example, canine heartworm treatment)? Again, is their data that can refute concerns that what you are seeing is prolonged antigen shedding from dead and decaying worms and not live adults?

Please address these and the other review comments in revision. If at all possible, I will attempt to review the revised manuscript without sending it out for further review in order to minimise any further delays.

Thank you

Aaron

In addition, I noted that you record in your supplementaries that you see a major decline in the estimated

We cannot make any decision about publication until we have seen the revised manuscript and your response to the reviewers' comments. Your revised manuscript is also likely to be sent to reviewers for further evaluation.

Sincerely,

Aaron R. Jex

Section Editor

Aaron Jex

Section Editor

Dear Dr Hoekstra,

Firstly, I'd like to apologise for the lengthy delay in processing your manuscript. I trust the journal has been in contact with you. This delay was in part due to, I think, major changes in the journals submission cite, which meant the article sat in limbo for some weeks before it actually appeared in my assignment feed. I then had some further significant delays with finding reviewers. I mention this not to make excuses for these delay but by way of acknowledging that the length of time it has taken to process your submission is not acceptable and I hope it will not be repeated in the future.

As you'll see from the reviewers comments, the reviews are quite split on your manuscript. Unfortunately, somehow, one of the reviewer's (who recommended rejection of the manuscript) comments were not uploaded or otherwise went missing. I can't say what has caused this, but I followed up with the review and they provided the following summary of their concerns.

"The manuscript "Limited efficacy of repeated praziquantel treatment in Schistosoma mansoni infections as revealed by highly accurate diagnostics, PCR and UCP-LF CAA (RePST trial)" aimed to evaluate PZQ efficacy using PCR and UCP-LF CAA in addition to the techniques already employed in this kind of study (KK and POC-CCA). However, the results did not show the relevance of the different methods used. Also, there is no relevant difference between the obtained data from KK and POC-CCA and PCR and UCP-LF CAA. Accordingly, I recommend rejecting the manuscript."

In an effort to prevent any further delays to processing your submission, I have informally reviewed the article myself, particularly in order to see if I could understand this second reviewer's concerns. I think it would be helpful to see the comparisons you undertook in your supplementaries featuring more prominently in your manuscript in order to address these concerns. I certainly won't tell you how to design your study or author your manuscript, but I think it might be helpful if you considered a main text figure and/or table (both perhaps?) clearly demonstrating:

1. The differences in sensitivity of the four methods you employed and their relative agreement with your surrogate 'gold-standard'. You mention in your text that the UPC-LC CAA method has been shown in prior study to have 'high' specificity and sensitivity, but I think demonstrating this using your control material is important for reader's in interpreting the PZQ CR results. I understand that this is in your supplementaries, but I think it might be better placed in the main text.

2. Similarly, you demonstrate in the supplementaries, as notable difference in the estimated CR based on the four methods you have used, but again, I think having this as a main figure and highlighting which of these differences are statistically significant would be helpful to the reader.

2. Finally, do you have data for the relative diagnostic specificity of each method? Particularly specificity for patent/live infections. With the antigen tests, is there any way to know for how long hosts will continue to shed worm antigens after the worms themselves have been killed through PZQ treatment? Similarly, is there any data that following the apparent cessation of egg production after PZQ (inferred by KK for example) that hosts still showing infection based on UPC-LC CAA or POC-CCA eventually resume egg-shedding (as evidence of the survival of any remaining worms and their return to reproductive fecundity). I presume such data would have to come from a hamster or similar model animal. I think this is particularly worth considering in the intense treatment group in whom you measured a substantial reduction in infection based on KK, but much less so with the other methods after week 2 or so. Could it be that either treatment has sterilised the surviving adults (as has been seen in, for example, canine heartworm treatment)? Again, is their data that can refute concerns that what you are seeing is prolonged antigen shedding from dead and decaying worms and not live adults?

Please address these and the other review comments in revision. If at all possible, I will attempt to review the revised manuscript without sending it out for further review in order to minimise any further delays.

Thank you

Aaron

In addition, I noted that you record in your supplementaries that you see a major decline in the estimated

Reviewer's Responses to Questions

**Key Review Criteria Required for Acceptance?**

**Methods**

-Are the objectives of the study clearly articulated with a clear testable hypothesis stated?

-Is the study design appropriate to address the stated objectives?

-Is the population clearly described and appropriate for the hypothesis being tested?

-Is the sample size sufficient to ensure adequate power to address the hypothesis being tested?

-Were correct statistical analysis used to support conclusions?

-Are there concerns about ethical or regulatory requirements being met?

Reviewer #1: (No Response)

Reviewer #2: Yes, the proposed objectives are in accordance with the tested hypothesis. The study design is in accordance with the proposed objective. The sample used here originates from a previous publication and although it started from a very large group of schoolchildren and only a little more than 125 children were left for the analyses, even so this number is enough to carry out the statistical analyzes which support well the conclusions drawn. 

The authors present the necessary approvals to the local ethics committee and the institution responsible for carrying out the exams.

**Results**

-Does the analysis presented match the analysis plan?

-Are the results clearly and completely presented?

-Are the figures (Tables, Images) of sufficient quality for clarity?

Reviewer #1: (No Response)

Reviewer #2: The results are well described and in accordance with the proposed objectives. The amount of tables and figures is enough to clearly show what is proposed. Although there is no gold standard for comparison, the authors determined for comparison the composite reference standard and considered the individual who was positive in the four tests truly positive.

**Conclusions**

-Are the conclusions supported by the data presented?

-Are the limitations of analysis clearly described?

-Do the authors discuss how these data can be helpful to advance our understanding of the topic under study?

-Is public health relevance addressed?

Reviewer #1: (No Response)

Reviewer #2: Yes, the conclusions are supported by the data presented. The data obtained make a great contribution to the question of eliminating schistosomiasis as proposed in the WHO guidelines. It is clear that the determination of the cure rate requires a more complex analysis. More sensitive methods certainly showed if, in fact, these worms were truly eliminated or not. It remained a curiosity to know if the analyzes had been extended for another 2 weeks how this data would behave. I consider that these data are relevant for Public Health.

**Editorial and Data Presentation Modifications?**

Reviewer #1: (No Response)

Reviewer #2: Although the authors refer to previous works, it would be important to make reference in the discussion (or even in the methodology) about the issue of rejection or even the difficulty of treating children with PZQ pills, speak also if in the intensive treatment group how they dealt with the adverse reactions that routinely occur. Another curiosity is to know if with so many evaluations (sample collections) there was no loss of individuals for not delivering the samples. A comment would be in order. They are minor modifications.

**Summary and General Comments**

Reviewer #1: (No Response)

Reviewer #2: The data presented clearly warns us that we need to rethink the treatment of schistosomiasis, not only in terms of the dose to be applied, but also in the search for new drugs.

PLOS authors have the option to publish the peer review history of their article (what does this mean?). If published, this will include your full peer review and any attached files.

Reviewer #1: No

Reviewer #2: No
---

## [Editor Report · Decision Letter 1]

6 Dec 2022

Dear Mrs Hoekstra,

We are pleased to inform you that your manuscript 'Limited efficacy of repeated praziquantel treatment in Schistosoma mansoni infections as revealed by highly accurate diagnostics, PCR and UCP-LF CAA (RePST trial)' has been provisionally accepted for publication in PLOS Neglected Tropical Diseases.

Best regards,

Aaron R. Jex

Section Editor

Aaron Jex

Section Editor

Thank you to the authors' for their patience with this submision and again my apologies for the delays during review. My apologies also for the confusion with my review. I got confused when reading the intial submission and had the impression that part of the intent was to demonstrate that the worm-based antigen methods were more sensitive etc that egg-based (ie, prooving the method) rather than that this was already well established and rather the intent was to show that applying these methods provides a more accurate representation of the efficacy of a PZQ trial result. In reading through the authors' responses and revisiting the manuscript, I'm not sure how I got this confused. Nonetheless, I thank the authors' for their responses and I feel this is clear to me now. Regarding my comment about the reduction in the differences in infection reductions as inferred by egg or worm-based approaches, again, my apologies for my imprecise language. I was referring to the data shown in Figure 1 (% positives after treatment) in relation to my question about what remaining antigen was either dying worms or worms rendered senescent by treatment. I thank the authors for clarifying this in their response.

---

## [Editor Report · Acceptance letter]

19 Dec 2022

Dear Mrs Hoekstra,

We are delighted to inform you that your manuscript, " Limited efficacy of repeated praziquantel treatment in Schistosoma mansoni infections as revealed by highly accurate diagnostics, PCR and UCP-LF CAA (RePST trial) ," has been formally accepted for publication in PLOS Neglected Tropical Diseases.

Best regards,

Shaden Kamhawi

co-Editor-in-Chief

Paul Brindley

co-Editor-in-Chief
